# Continuous positive airway pressure to reduce the risk of early peripheral oxygen desaturation after onset of apnoea in children: A double-blind randomised controlled trial

Jayme Marques dos Santos Neto[1]☉*, Clístenes Cristian de Carvalho[2,3‡], Lívia Barboza de Andrade[2‡], Thiago Gadelha Batista Dos Santos[1‡], Rebeca Gonelli Albanez da Cunha Andrade[2‡], Raphaella Amanda Maria Leite Fernandes[2‡], Flavia Augusta de Orange[1,2☉]

1 Support and Therapeutic Diagnosis Division, Anesthesiology and Post-Anesthetic Care Unit, Federal University of Pernambuco's Teaching Hospital, Recife, Pernambuco, Brazil, 2 Department of Post-graduation, *Instituto de Medicina Integral Prof. Fernando Figueira*, Recife, Pernambuco, Brazil, 3 Department of Surgery, Federal University of Campina Grande, Campina Grande, Paraíba, Brazil

☉ These authors contributed equally to this work.
‡ These authors also contributed equally to this work.
* jaymemed@yahoo.com.br

## Abstract

Continuous positive airway pressure (CPAP) during anaesthesia induction improves oxygen saturation ($SpO_2$) outcomes in adults subjected to airway manipulation, and could similarly support oxygenation in children. We evaluated whether CPAP ventilation and passive CPAP oxygenation in children would defer a $SpO_2$ decrease to 95% after apnoea onset compared to the regular technique in which no positive airway pressure is applied. In this double-blind, parallel, randomised controlled clinical trial, 68 children aged 2–6 years with ASA I–II who underwent surgery under general anaesthesia were divided into CPAP and control groups (n = 34 in each group). The intervention was CPAP ventilation and passive CPAP oxygenation using an anaesthesia workstation. The primary outcome was the elapsed time until $SpO_2$ decreased to 95% during a follow-up period of 300 s from apnoea onset (T1). We also recorded the time required to regain baseline levels from an $SpO_2$ of 95% aided by positive pressure ventilation (T2). The median T1 was 278 s (95% confidence interval [CI]: 188–368) in the CPAP group and 124 s (95% CI: 92–157) in the control group (median difference: 154 s; 95% CI: 58–249; p = 0.002). There were 17 (50%) and 32 (94.1%) primary events in the CPAP and control groups, respectively. The hazard ratio was 0.26 (95% CI: 0.14–0.48; p<0.001). The median for T2 was 21 s (95% CI: 13–29) and 29 s (95% CI: 22–36) in the CPAP and control groups, respectively (median difference: 8 s; 95% CI: -3 to 19; p = 0.142). $SpO_2$ was significantly higher in the CPAP group than in the control group throughout the consecutive measures between 60 and 210 s (with p ranging from 0.047 to <0.001). Thus, in the age groups examined, CPAP ventilation and passive CPAP

**Data Availability Statement:** All relevant data are within the manuscript and its Supporting Information files.

**Funding:** JMSN received a scholarship provided by Coordenação de Aperfeiçoamento de Pessoal de Nível Superior (CAPES) (https://www.gov.br/capes/pt-br). The funders had no role in study design, data collection and analysis, decision to publish, or preparation of the manuscript.

**Competing interests:** The authors have declared that no competing interests exist.

oxygenation deferred $SpO_2$ decrease after apnoea onset compared to the regular technique with no positive airway pressure.

## Introduction

General anaesthesia largely alters the physiology of the respiratory system. Effects such as impairment of functional residual capacity (FRC) and pulmonary compliance, emergence of atelectasis, and disturbances in the ventilation–perfusion ratio (V/Q) increase the chances of hypoxaemia [1–5].

Paediatric patients have the highest risk of desaturation during anaesthesia induction [5–8], particularly because of their physiological characteristics (lower FRC and greater oxygen consumption). Additionally, their anatomy (proportionally large head and tongue, tonsil and adenoid hypertrophy, small and narrow hypopharynx, upper larynx, slanted vocal cords, U-shaped inverted epiglottis, and short airway radius) may complicate airway management. Altogether, these factors contribute to triggering and maintaining hypoxaemia [7,9–11], rendering children more vulnerable to severe complications, such as cardiorespiratory arrest and death [7].

Pre-oxygenation lengthens safe apnoea time and is often used to prevent hypoxaemia [7,9,12]. Despite its benefits, pre-oxygenation, particularly with pure oxygen, can contribute to the occurrence of micro-atelectasis and changes in the V/Q during anaesthesia induction. Alveolar recruitment manoeuvres and the use of positive end-expiratory pressure can reverse and prevent these events [4,5]. Other preventive strategies have also been evaluated; however, an ideal technique has not been determined thus far [13,14].

Continuous positive airway pressure (CPAP) allows patients to breathe spontaneously through a pressurised circuit, thus improving alveolar gas exchange, reducing atelectasis incidence, increasing FRC and tidal volume, and reversing the anaesthetic-induced upper airway cross-sectional area reduction [15,16]. Its use to facilitate the treatment of patients with diseases like obstructive sleep apnoea is well documented and recommended in the perioperative period [17]. In paediatrics, the application of CPAP has been widely studied in patients with bronchiolitis as an alternative to controlled mechanical ventilation [18,19]. Although the evidence for its use during anaesthesia induction in children is limited, the results obtained in the adult population are encouraging [20–24].

Therefore, we aimed to evaluate whether CPAP ventilation and passive CPAP oxygenation would defer oxygen saturation ($SpO_2$) decrease to 95% after apnoea onset in children compared to the regular technique in which no positive airway pressure is delivered. Additionally, we measured the $SpO_2$ values and recorded the time until recovery of $SpO_2$ to pre-apnoea levels.

## Materials and methods

We conducted a double-blind, parallel, randomised controlled clinical trial in children who underwent surgery at the Federal University, Pernambuco's Teaching Hospital between March 2018 and May 2019. This study was approved by the Ethical Committee of *Instituto de Medicina Integral Prof. Fernando Figueira*, Recife, Brazil (Approval No. CAAE: 79591417.0.0000.5201) on 27 December 2017 and was registered at ClinicalTrials.gov in February 2018 (NCT03432390). Written informed consent was obtained from the parents or legal

guardians of all participants (Study protocol https://dx.doi.org/10.17504/protocols.io.bqv5mw86).

Children aged 2–6 years, with an ASA status of I or II, who underwent elective surgery under general anaesthesia were included in the study. The exclusion criteria were pre-existing parenchymal lung disease, cyanosis or $SpO_2$ <95% prior to anaesthesia induction, and a current upper respiratory tract infection or a history of it in the preceding 4 weeks.

The intervention consisted of CPAP ventilation and passive CPAP oxygenation using the anaesthesia workstation, whereas the main outcome was the elapsed time until $SpO_2$ decreased to 95% in each patient during the 5-min follow-up after apnoea onset (T1).

Meanwhile, we recorded $SpO_2$ values. We also noted the time required for recovery of $SpO_2$ from 95% to pre-apnoea levels (T2).

Likewise, patients in the control group ventilated in the same manner in the anaesthesia workstation without adding positive pressure to the circuit.

Participants were recruited from the surgical ward by a research assistant who was blinded to the group allocation of the patient.

The patients were randomised to either the CPAP or control groups. The random allocation sequence was generated by JMSN using the Random Allocation Software program (Version 1.0) with numbers generated in blocks of eight.

Sequentially numbered, otherwise identical, sealed envelopes, each containing a 2-inch by 2-inch piece of paper with a written code indicating the intervention group or the control group were used for each child to ensure allocation concealment. The envelopes were opaque and sequentially assigned to each new patient. Just before a patient's admission to the operating room, the attending anaesthesiologist, who was not part of the study team, took note of the allocation group, keeping those involved in the study unaware of the assigned intervention. The attending anaesthesiologist was then required to arrange the setting in order to keep the allocation concealed and ensure protocol compliance.

When CPAP was used, the adjustable pressure-limiting (APL) valve was set to provide a pressure of 10 $cmH_2O$. In the control group, the APL valve was left open, thus maintaining a pressure of 0 $cmH_20$. Thereafter, the valve was covered by a surgical sheet so that the allocation remained concealed and no one, other than the attending anaesthesiologist, knew the intervention assigned. Whether or not the patient was premedicated depended on the discretion of the attending anaesthesiologist.

The patient and researchers were thereafter allowed to access the operating room. Routine monitoring was performed with continuous eletrocardiogram, pulse oximeter placed on the patient's finger, non-invasive blood pressure assessment, and capnography. Inhalation induction of anaesthesia was then performed by JMSN, TGBS, or FAO, who were blinded to the intervention, using a facemask connected to the circle system of the anaesthesia workstation (Carestation 620, Datex-Ohmeda, Inc. Madison, WI) and attached to the patient using an elastic strap. Sevoflurane (8%) along with a 60% fraction of oxygen under a fresh gas flow of 4 L.$min^{-1}$ (2 L.$min^{-1}$ of oxygen/2 L.$min^{-1}$ of air) was initially provided to patients in both groups until the loss of the eyelid reflex. Thereafter, the concentration of sevoflurane was reduced to 4%. The patients remained on spontaneous ventilation. To assess their competence, we evaluated the capnography waveform, $ETCO_2$, abdominal movement, and chest expansion. At this point, a peripheral vein was cannulated for hydration and a standardised regimen of 3.5 mg.$kg^{-1}$ propofol bolus was injected to induce apnoea. All patients were ventilated for an equal period before the start of apnoea.

T1 was timed using an iPhone 7 Plus 12.3.1 (Apple Inc., Cupertino, CA) beginning from the onset of apnoea, identified as an absence of both respiratory movements and a capnography trace. When a period of 300 s was completed without reaching 95% $SpO_2$, the

measurement was halted and manually assisted lung ventilation was initiated. During the T1 recording, $SpO_2$ was noted. Once 95% $SpO_2$ was reached or 300 s passed, the attending anaesthesiologist covertly manipulated the APL valve, either truly or falsely, to maintain allocation concealment. When the valve was open, a pressure of 10 $cmH_2O$ was set. This way, regardless of the initial group to which the patient was allocated, from this moment onwards, the patient received assisted lung ventilation supported by a pressure of 10 $cmH_2O$. At that time, T2 was measured for those who reached 95% $SpO_2$.

Any adverse event that occurred during the study period was recorded.

A research assistant, the outcome assessor, blinded to patient allocation, performed all data collection.

## Statistical analysis

Under the proportional hazards assumption, which we validated using a test based on Schoenfeld residuals, we estimated that with a sample of at least 33 patients in each randomised group, the study would have an 80% power to detect a 50% reduction in the hazard of the CPAP group, allowing only a 5% chance of a type I error in a two-sided significance test. However, to compensate for any losses following randomisation (predicted at approximately 10%), this number was increased to 72 patients, with 36 in each group. The sample size was calculated using Schoenfeld's procedure. It was based on the a priori calculation, but a posteriori a second analysis was performed based on feedback that the initial assumptions required adjustment, and this analysis revealed that the sample size should be 76. Data analysis was performed using STATA version 12.1 SE (StataCorp, College Station, TX, USA). Descriptive statistical analysis was performed using measures of central tendency and dispersion for the quantitative variables and frequency distribution for the qualitative variables. For continuous measures, results are reported as means and SDs for normally distributed variables and medians and IQRs for non-normal quantities. A standard risk of 1.0 was attributed to the reference category. Survival analysis was conducted on the survival times to an $SpO_2$ of 95%. Survival probabilities were calculated using the Kaplan–Meier method, and the survival curves were compared using the log-rank test. The association measure for the primary outcome was the hazard risk ratio and, for the secondary endpoints, the median time difference. The medians of survival time in each group were estimated and compared using Laplace regression. Furthermore, a curve was constructed with the mean $SpO_2$ measurements at 30-s intervals by adjusting a linear regression model for correlated data that also evaluated the significance of time, group, and the interaction between them. The means were compared between the groups at each measured time using the Wald test, and the Benjamini–Hochberg procedure was applied to adjust the p-value to avoid type I error rate inflation. Statistical significance was defined as $p<0.05$. To test the robustness of the study, we calculated its fragility index using Fisher's exact test.

## Results

Of the 98 patients screened, 72 were included in the study. Four patients were lost to follow-up (n = 2 in each group) due to changes in anaesthetic induction protocol (2), absence of apnoea (1), and surgery suspension (1). Therefore, the final analysis included data from 68 patients for the primary outcome (n = 34 in each group) according to the CONSORT flow diagram (Fig 1). Both groups were similar in terms of physical and clinical characteristics and types of surgery performed (Table 1).

The median for T1 was 278 s (95% CI: 188–368) in the CPAP group and 124 s (95% CI: 92–157) in the control group (median difference: 154 s; 95% CI: 58–249; p = 0.002). The least time

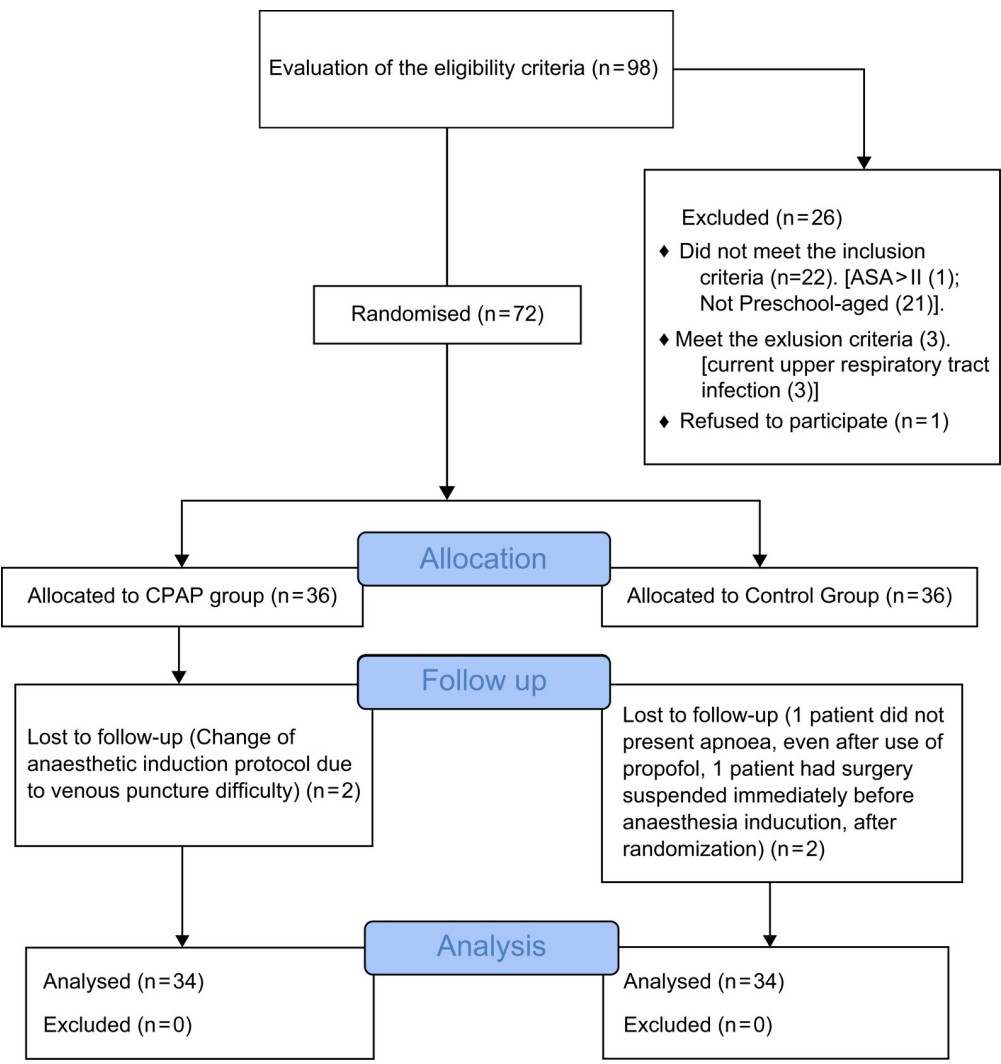

**Fig 1. CONSORT diagram of patient recruitment.**

to reach an SpO$_2$ of 95% was 75 and 30 s in the CPAP and control groups, respectively (Table 2).

Fig 2 shows the Kaplan-Meier curves for the CPAP and control groups. The log-rank test showed a p-value of <0.001. The estimated 300-s cumulative primary event rates were 50% in the CPAP group and 94.1% in the control group (proportional-hazards risk ratio: 0.26; 95% CI: 0.14 to 0.48; p<0.001).

The median for T2 was 21 s (95% CI: 13–29; minimum–maximum values: 10–170) in the CPAP group and 29 s (95% CI: 22–36; minimum–maximum values: 10–360) in the control group, with a non-significant between-group difference (median difference: 8 s; 95% CI: -3 to 19; p = 0.142) (Table 2).

The repeated-measures saturation values were higher in the CPAP group with a significant between groups difference in the 60–210 s interval (Fig 3).

No adverse events of a specific type and severity, such as laryngospasm, bronchospasm, bradycardia, cardiac arrest, or death, occurred in either group.

**Table 1. Characteristics of children with CPAP-ventilation and passive CPAP-oxygenation (10 cmH$_2$O) or no positive pressure (0 cmH$_2$O) during anaesthesia induction for elective surgery.**

|  | Group | |
|---|---|---|
|  | **CPAP (n = 34)** | **Control (n = 34)** |
| **Sex** | | |
| Female | 6 (17.7) | 10 (29.4) |
| Male | 28 (82.3) | 24 (70.6) |
| **Age, years** | 4.3 (1.4) | 4.2 (1.51) |
| **Weight, kg** | 18.3 (4.9) | 17.8 (4.58) |
| **ASA** | | |
| I | 31 (91.2) | 33 (97) |
| II | 3 (8.8) | 1 (3) |
| **Type of surgery** | | |
| Urological surgery | 15 (44.1) | 11 (32.3) |
| Herniorrhaphy | 12 (35.3) | 15 (44.1) |
| Combined surgery | 3 (8.8) | 3 (8.8) |
| Intraperitoneal surgery | 2 (5.9) | 1 (2.9) |
| Soft tissue tumour | 2 (5.9) | 4 (11.9) |

ASA, American Society of Anesthesiologists; CPAP, continuous positive airway pressure. Values are mean (SD) or number (proportion).

The study's Fragility Index was 9, meaning that it would take nine patients in the CPAP group to have their status change from non-events to events for the between-group difference to shift to being not statistically significant.

## Discussion

In this study, CPAP ventilation and passive CPAP oxygenation deferred SpO$_2$ decrease to 95% after apnoea onset compared to the regular technique in which no positive airway pressure was applied. Patients took more time to reach an SpO$_2$ of 95% in the CPAP group than in the control group. In the CPAP group, patients presented a 50% probability of reaching an SpO$_2$ of 95% within 5 min; in the control group, this probability increased to 94.1%. In addition, the risk of desaturation to SpO$_2$ in the CPAP group was approximately a quarter of that in the control group. Therefore, as swift tracheal intubation is not always possible, mainly for difficult airways, the assurance of optimal SpO$_2$ for longer periods would be of great value to enhance patient safety. As children have an increased basal metabolic rate, once an imbalance between oxygen supply and consumption arises, the final result is an overall progressive decrease in

**Table 2. T1 and T2 in children treated with CPAP ventilation and passive CPAP oxygenation (10 cmH$_2$O) or no positive airway pressure (0 cmH$_2$O) during anaesthesia induction for elective surgery.**

|  | Group | | Median difference | p-value |
|---|---|---|---|---|
|  | **CPAP** | **Control** |  |  |
| **T1, s** | 278 (188–368) | 124 (92–157) | 154 (58–249) | 0.002 |
| **T2, s** | 21 (13–29) | 29 (22–36) | 8 (-3 to 19) | 0.142 |

T1, time between apnoea onset and an SpO$_2$ of 95% during a maximum observation time of 300 s; T2, time from an SpO$_2$ of 95% until recovery to pre-apnoea levels; SpO$_2$, pulse oximetry oxygen saturation; CPAP, continuous positive airway pressure. Values are presented as median (95% CI).

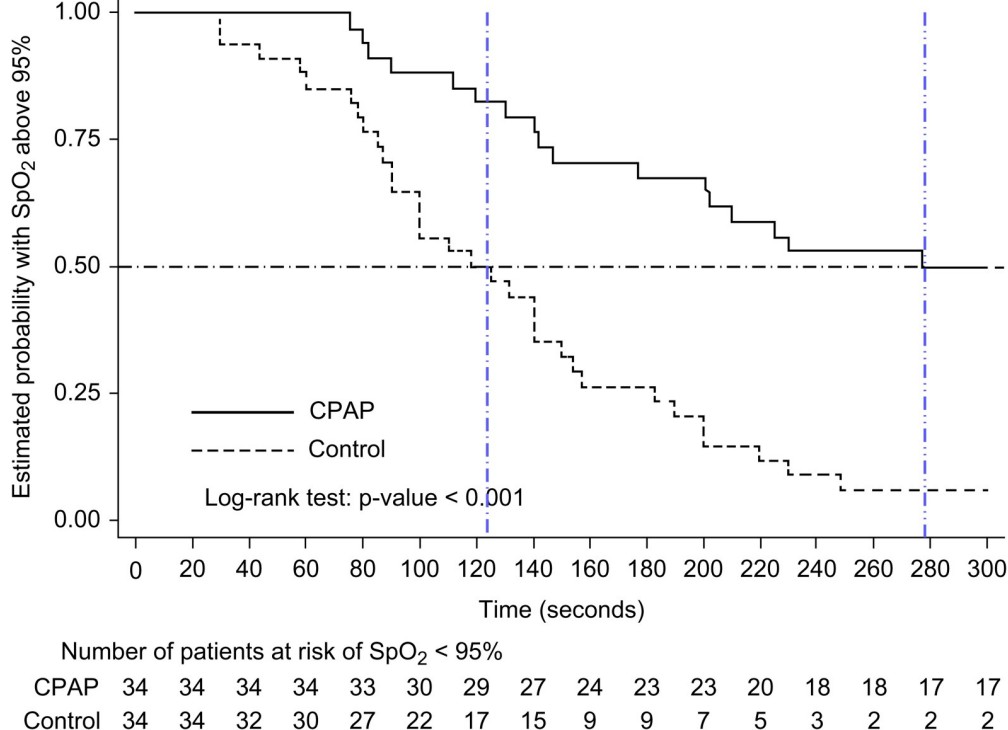

Number of patients at risk of SpO₂ < 95%

| | | | | | | | | | | | | | | | |
|---|---|---|---|---|---|---|---|---|---|---|---|---|---|---|---|
| CPAP | 34 | 34 | 34 | 34 | 33 | 30 | 29 | 27 | 24 | 23 | 23 | 20 | 18 | 18 | 17 | 17 |
| Control | 34 | 34 | 32 | 30 | 27 | 22 | 17 | 15 | 9 | 9 | 7 | 5 | 3 | 2 | 2 | 2 |

**Fig 2. Kaplan–Meier curves for the CPAP and control groups.** Occurrence of an SpO₂ of 95% during a 5-min follow-up in children with CPAP ventilation and passive CPAP oxygenation (10 cmH₂O) or no positive airway pressure (0 cmH₂O) during anaesthesia induction for elective surgery showing a significant difference between the survival curves (log-rank test; p<0.001).

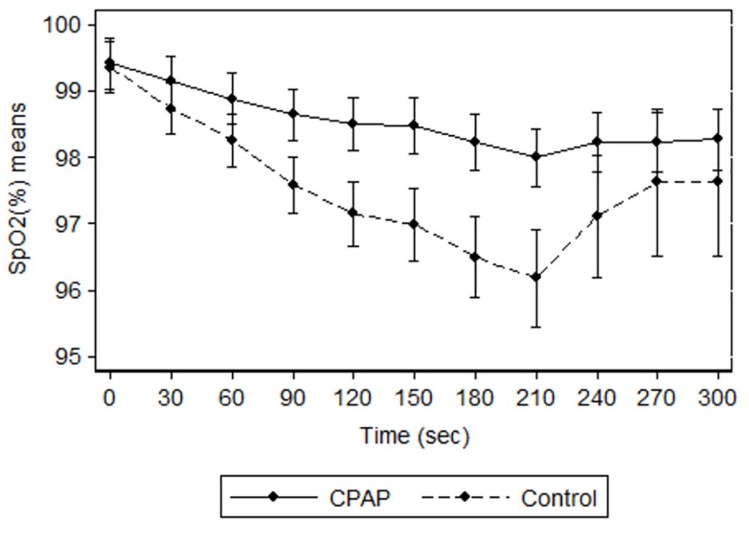

**Fig 3. Repeated-measures saturation values curves.** Mean SpO₂ values in children with either CPAP ventilation and passive CPAP oxygenation (10 cmH₂O) or no positive airway pressure (0 cmH₂O) during anaesthesia induction for elective surgery and their respective confidence intervals in the two-group interaction (p ranging from 0.047 to <0.001 for the interaction between the curves in the 60–210 s interval).

oxygen levels (hypoxic hypoxia). Accordingly, CPAP could possibly play a critical role in preventing the occurrence of secondary severe complications due to oxygen desaturation, consistent with findings of the Paediatric Difficult Intubation Registry, which showed that such complications were more frequent in patients with unanticipated difficult airways [7]. Thus, in clinical practice, CPAP would be useful in situations where apnoea is either expected to occur (such as during anaesthesia induction) or required (e.g., during magnetic resonance imaging).

The shortest interval spent to reach an SpO2 of 95% in the CPAP group was 75 s, whereas in the control group it was 30 s. In other words, for desaturation to be avoided in our cohort, the airway should be defined within 30 s for patients with no pressure support, and within 75 s for those given the support.

To better assess the reliability of our results, we calculated the fragility index (FI = 9) for the association found, which provided further support for our findings and ensured a low chance of random error. In other words, it would take nine non-event patients in the CPAP group for their status to change to having events for statistical significance to be lost. This number exceeds the number of patients lost to follow-up. Moreover, FI has been used in clinical trials with dichotomous outcomes to appraise its robustness and to complement the interpretation of the P-value. Although there is no cut-off, the greater the FI value, the greater the robustness of the study's results [25].

This may be the first study evaluating the role of CPAP in paediatric patients during anaesthesia induction. However, despite the shortage of investigations in children, there have been many studies that have evaluated the application of positive airway pressure in adult populations during preoxygenation and anaesthesia induction [20–24]. Their results point mostly in the same direction as ours, showing improvements in outcomes, such as longer duration of apnoea before clinically significant arterial desaturation, better oxygenation and prevention of desaturation episodes, upon the use of positive airway pressure. These authors attributed the improved outcomes to the increase in FRC (associated with reduction of shunt areas and improvement in the V/Q). Presumably, the most suitable means for the anaesthesiologist to manage situations where apnoea is expected to occur is to apply techniques that increase lung oxygen reserves. This is also our main hypothesis, although a CT scan-based study carried out in children was unable to confirm the association between CPAP-assisted oxygenation and the smallest areas of atelectasis [26]. Thus, further studies are necessary to better understand the basal mechanisms involved in CPAP benefits.

Two randomised clinical trials in adults have demonstrated results similar to ours. The first study in obese patients demonstrated extended nonhypoxic apnoea time when applying pressure support, using a combination of CPAP and PEEP, compared to the regular technique in which no positive airway pressure support was applied [23,24]. The second study, also in adults, with methods similar to ours, applied CPAP during both preoxygenation and a given period of apnoea. It also demonstrated a slowed desaturation [22]. Consistent with these findings, the use of CPAP—5 cmH$_2$O through a Mapleson A circuit, only during preoxygenation, was likewise shown to defer desaturation in adults [20]. Thus, the use of positive airway pressure during anaesthesia induction to improve respiratory outcomes has been demonstrated to succeed in many different scenarios.

As previously stated, we expected CPAP to prevent atelectasis, preserving the FRC, and achieving the improved outcomes we observed. This technique is assumed to provide more oxygen during apnoea by increasing the transpulmonary pressure secondary to a continuous flow of gases, which delays the occurrence of hypoxaemia [27]. Likewise, CPAP also manages to preserve the upper airway's cross-sectional area. Altogether, these effects improve both spontaneous ventilation and patient lung volumes, and are in accordance with our findings.

To reduce the use of pure oxygen and minimise the likelihood of absorption atelectasis due to denitrogenation, we applied an inspired oxygen fraction of 60% [4,28]. Thus, a combination of lower fractions of oxygen and improved FRC, achieved by the use of positive airway pressure, may constitute a more effective and less harmful way of obtaining satisfactory pre-oxygenation.

Regarding the limit of 95% set as our endpoint for $SpO_2$, although lower levels are usually reported in studies about safe apnoea time [13,20,29], we chose this threshold to ensure patient safety. Since a decrease in $SpO_2$ below that level unsettles heart function and haemodynamic parameters such as the systolic index [30], and because we did not aim to document such changes, we avoided submitting the participants to these threats. Hence, we were also convinced to limit the observation time to 5 min due to ethical issues discussed with members of our hospital's anaesthesiology clinical staff.

Here, CPAP failed to shorten the time to restore the previous levels of $SpO_2$. Similarly, another clinical trial also did not demonstrate statistically significant differences when analysing the same outcome in apnoeic adults after pre-oxygenation with and without CPAP 5 $cmH_2O$. For this study, the authors started timing from an $SpO_2$ of 93% [20]. However, for this outcome, our results may be considered underpowered as the sample-size estimation did not take this parameter into account, which may have led us to a type 2 error.

Further grounding the benefit of early CPAP application during anaesthesia is the progress of the measured $SpO_2$. Oxygen saturation was consistently higher in the CPAP group during the period between 60 and 210 s (p ranging from 0.047 to <0.001). Different clinical trials in diverse scenarios have likewise presented CPAP to be firmly associated with the highest arterial oxygen pressure [20–24]. However, we were not able to find significant between-group differences for up to 1 min or > 210 s. Studies powered for these specific periods of time might better investigate such associations.

Despite the relevant statistical results obtained, this was a small, single-centre study with a narrow population age range; thus, further studies are necessary to establish the benefits demonstrated in this study. The results may be underpowered as highlighted during the a posteriori adjustment made in the sample size calculation. The actual study sample was 68 patients. With this sample size we achieved 72,4% power to detect a 50% reduction in the hazard of the experimental group, with a 5% significance level. We did not record expired oxygen and nitrogen, the amount of $CO_2$ expired in the return of pulmonary ventilation, or the stomach size during induction. In particular, the pressure used was 10 $cmH_2O$, which was below the values related to stomach enlargement in anaesthetized children [31,32]. Although the APL valve was covered with a surgical sheet, the researchers had access to each patient's saturation and apnoea time, which may have potentially compromised the blinding process. We did not address the prevalence of obstructive sleep apnoea or calculate body mass index; therefore, it was not possible to perform subgroup analyses. Furthermore, the CPAP group received positive airway pressure during both spontaneous ventilation and apnoea, whereas the control group did not. As patients who received CPAP would have received a degree of apnoeic oxygenation, halting positive airway pressure from apnoea onset might be more informative for clinical practice.

## Conclusions

CPAP ventilation and passive CPAP oxygenation deferred $SpO_2$ decrease after apnoea onset in the age groups examined in this study compared to the regular technique in which no positive airway pressure was used. The CPAP technique was easy to apply and appeared to enhance patient safety.

## Supporting information

**S1 Fig. CONSORT diagram of patient recruitment.**
(TIF)

**S2 Fig. Kaplan–Meier curves for the CPAP and control groups.** Occurrence of an $SpO_2$ of 95% during a 5-min follow-up in children with CPAP ventilation and passive CPAP oxygenation (10 $cmH_2O$) or no positive airway pressure (0 $cmH_2O$) during anaesthesia induction for elective surgery showing a significant difference between the survival curves (log-rank test; $p < 0.001$).
(TIF)

**S3 Fig. Repeated-measures saturation values curves.** Mean $SpO_2$ values in children with either CPAP ventilation and passive CPAP oxygenation (10 $cmH_2O$) or no positive airway pressure (0 $cmH_2O$) during anaesthesia induction for elective surgery and their respective confidence intervals in the two-group interaction (p ranging from 0.047 to <0.001 for the interaction between the curves in the 60–210 s interval).
(TIF)

**S1 File. CONSORT 2010 Checklist.**
(PDF)

**S2 File. Study Protocol in Portuguese.**
(PDF)

**S3 File. Study Protocol.**
(DOCX)

**S4 File. CPAP group data Time to desaturation.** Database containing each CPAP group patients time to a $SpO_2$ of 95% or 300 seconds.
(PDF)

**S5 File. Control group data Time to desaturation.** Database containing each Control group patients time to a $SpO_2$ of 95% or 300 seconds.
(PDF)

**S6 File. Repeated-measures saturation values.** Database containing each patient $SpO_2$ during apnoea time until $SpO_2$ of 95% or 300 seconds.
(PDF)

**S7 File. CPAP group data Time for recovery.** Database containing each CPAP group patients time required for recovery of $SpO_2$ from 95% to pre-apnoea levels (T2).
(PDF)

**S8 File. Control group data Time for recovery.** Database containing each Control group patients time required for recovery of $SpO_2$ from 95% to pre-apnoea levels (T2).
(PDF)

**S9 File. Data used to build repeated-measures saturation values curves.**
(PDF)

**S10 File. Database with raw data of all patients.**
(XLSX)

**S11 File.**
(PDF)

## Acknowledgments

The authors acknowledge the support received from the Teaching Hospital of the Federal University of Pernambuco, where patient data were collected, and *Instituto de Medicina Integral Prof. Fernando Figueira*, where the study was conceived. The authors are grateful to Professor José Natal Figueroa for his essential contribution to data analysis and interpretation, and to the medical residents in anaesthesiology, Raniere Nobre Fonseca, and Victor Lemos Macedo for their valuable collaboration in performing the data collection. We would like to thank Editage (www.editage.com) for English language editing.

## Author Contributions

**Conceptualization:** Jayme Marques dos Santos Neto, Lívia Barboza de Andrade, Rebeca Gonelli Albanez da Cunha Andrade, Flavia Augusta de Orange.

**Data curation:** Jayme Marques dos Santos Neto, Clístenes Cristian de Carvalho, Lívia Barboza de Andrade, Thiago Gadelha Batista Dos Santos, Flavia Augusta de Orange.

**Formal analysis:** Jayme Marques dos Santos Neto, Clístenes Cristian de Carvalho, Lívia Barboza de Andrade, Thiago Gadelha Batista Dos Santos, Rebeca Gonelli Albanez da Cunha Andrade, Raphaella Amanda Maria Leite Fernandes, Flavia Augusta de Orange.

**Investigation:** Jayme Marques dos Santos Neto, Thiago Gadelha Batista Dos Santos, Flavia Augusta de Orange.

**Methodology:** Jayme Marques dos Santos Neto, Clístenes Cristian de Carvalho, Lívia Barboza de Andrade, Rebeca Gonelli Albanez da Cunha Andrade, Flavia Augusta de Orange.

**Project administration:** Jayme Marques dos Santos Neto, Thiago Gadelha Batista Dos Santos, Flavia Augusta de Orange.

**Supervision:** Lívia Barboza de Andrade, Rebeca Gonelli Albanez da Cunha Andrade, Flavia Augusta de Orange.

**Validation:** Clístenes Cristian de Carvalho, Lívia Barboza de Andrade, Thiago Gadelha Batista Dos Santos, Rebeca Gonelli Albanez da Cunha Andrade, Raphaella Amanda Maria Leite Fernandes, Flavia Augusta de Orange.

**Writing – original draft:** Jayme Marques dos Santos Neto, Clístenes Cristian de Carvalho, Lívia Barboza de Andrade, Thiago Gadelha Batista Dos Santos, Rebeca Gonelli Albanez da Cunha Andrade, Raphaella Amanda Maria Leite Fernandes, Flavia Augusta de Orange.

**Writing – review & editing:** Jayme Marques dos Santos Neto, Clístenes Cristian de Carvalho, Lívia Barboza de Andrade, Thiago Gadelha Batista Dos Santos, Rebeca Gonelli Albanez da Cunha Andrade, Raphaella Amanda Maria Leite Fernandes, Flavia Augusta de Orange.

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
