## [Decision Letter · Decision Letter 0]

16 Dec 2020

PONE-D-20-33942

Continuous positive airway pressure to reduce the risk of early peripheral oxygen desaturation following apnoea onset in children: randomised double-blind controlled trial

PLOS ONE

Dear Dr. dos Santos Neto,

Thank you for submitting your manuscript to PLOS ONE. After careful consideration, we feel that it has merit but does not fully meet PLOS ONE’s publication criteria as it currently stands. Therefore, we invite you to submit a revised version of the manuscript that addresses the points raised during the review process.

Our reviewers, experts in the field, have provided valuable comments, which I share. Please consider them carefully in all points when you submit a revision of your paper.

We look forward to receiving your revised manuscript.

Kind regards,

Thomas Penzel

Academic Editor

PLOS ONE

Journal Requirements:

"The authors acknowledge the support received from the Teaching Hospital of the Federal

University of Pernambuco where patient data were collected and the Instituto de Medicina

Integral Prof. Fernando Figueira, in which the study was conceived. The authors are grateful to

Professor José Natal Figueroa for his essential contribution with data analysis and interpretation,

the medical residents in anaesthesiology, Raniere Nobre Fonseca and Victor Lemos Macedo, for

their valuable collaboration in performing the data collection and to the Coordenação de

Aperfeiçoamento de Pessoal de Nível Superior (CAPES) for providing the author with a

scholarship."

"The author(s) received no specific funding for this work"

Reviewers' comments:

Reviewer's Responses to Questions

**Comments to the Author**

1. Is the manuscript technically sound, and do the data support the conclusions?

Reviewer #1: Yes

Reviewer #2: Partly

2. Has the statistical analysis been performed appropriately and rigorously? 

Reviewer #1: Yes

Reviewer #2: No

3. Have the authors made all data underlying the findings in their manuscript fully available?

Reviewer #1: Yes

Reviewer #2: Yes

4. Is the manuscript presented in an intelligible fashion and written in standard English?

Reviewer #1: Yes

Reviewer #2: Yes

5. Review Comments to the Author

Reviewer #1: The authors aimed, with a randomized double-blind parallel controlled clinical trial, to evaluate whether CPAP-ventilation and passive CPAP oxygenation would reduce the risk of a SpO2 decrease to 95% following apnea onset in children as compared to no positive airway pressure. This trial was successfully accomplished in 68 children aging 2-6 years during anesthesia, and is commendable. The timeframe evaluated was 5 minutes after the start of an apnea, and recording was every 30 seconds. They found that while in the CPAP group, patients presented a 50% probability of having reached a SpO2 of 95% within 5 minutes, in the Control group this chance increased to 94.1%. In other words, a prosperous result.

Two minor comments are:

1. Could the authors elaborate on the stability and the severity of SpO2 in the 30-second intervals within the CPAP group? Namely, this analysis (and hence result) could be related back to the limitation of the spontaneous ventilation and apnea situation during their anesthesia.

2. Could the authors discuss the applicability of this approach to children with developmental disabilities undergoing surgery.

Reviewer #2: - the primary outcome is not clearly described. At first reading, it seems to be the risk of SpO2 of 95% within a 5 minute period (indeed the sample size calculation is that for estimating a risk/proportion). However, reading the manuscript it is repeatedly referred to as hazard of the outcome, implying that the outcome is a time-to-event. The authors need to clarify this.

- the units and resolution of measurement of the outcome T1 is not clear: was it measured down to the second or was it also measured in 30 second increments?

- if the primary outcome is a risk/proportion, then the sample size calculation is not sufficiently described as it does not include the 'baseline' risk of the outcome; this should be included along with a citation to support it. It is also not clear whether the 50% risk reduction proposed is an absolute or relative reduction. However, if the outcome is time-to-event, then the sample size calculation is incorrect, because other than including the hazard of the outcome in the 'baseline' group, it must also include some determination of the expected number of events.

- for the descriptive analysis the authors should follow standard reporting for randomised trials, where the mean and SD for continuous baseline variables or counts and proportions for categorical baseline variables be presented by treatment arm without statistical tests comparing the arms.

- if the primary outcome is a risk within a defined time period (5 minutes) therefore describing it as a hazard and estimating hazard ratios comparing the treatment arms for this outcome is not appropriate. It is however appropriate for the secondary outcome T1 (and perhaps T2).

- the results for the three outcomes should be tabulated in the standard reporting format for a clinical trial i.e. including the event numbers and denominators per group, risks/hazards/rates as appropriate and their ratios, confidence intervals and p-values.

- please indicate whether the trial protocol submitted with this manuscript was published before the analysis was undertaken, as I am not able to determine that.

6. PLOS authors have the option to publish the peer review history of their article (what does this mean?). If published, this will include your full peer review and any attached files.

Reviewer #1: No

Reviewer #2: No

---

## [Author Response · Author response to Decision Letter 0]

28 Jan 2021

Please ensure that your manuscript meets PLOS ONE's style requirements, including those for file naming. Adjusted in the manuscript.

We suggest you thoroughly copyedit your manuscript for language usage, spelling, and grammar. If you do not know anyone who can help you do this, you may wish to consider employing a professional scientific editing service. The original manuscript with tracked changes has been sent to Editage website (www.editage.com). Some changes in the text has been made before and after the edition. Due to this, the Revised Manuscript with Track Changes is different from the Manuscript.

Please remove any funding-related text from the manuscript and let us know how you would like to update your Funding Statement. Please include your amended statements within your cover letter. Funding-related text removed from the manuscript. JMSN received a scholarship provided by Coordenação de Aperfeiçoamento de Pessoal de Nível Superior (CAPES) (https://www.gov.br/capes/pt-br). The funders had no role in study design, data collection and analysis, decision to publish, or preparation of the manuscript.

Please include a separate caption for each figure in your manuscript. Adjusted in the manuscript.

Could the authors elaborate on the stability and the severity of SpO2 in the 30-second intervals within the CPAP group? Namely, this analysis (and hence result) could be related back to the limitation of the spontaneous ventilation and apnea situation during their anesthesia. Thank you. The curve of repeated measures shows a less pronounced drop compared to that of the control group. We can theorize that this is due to greater alveolar stability with maintenance of the opening of these and preservation of the functional residual capacity, the main body oxygen supply. During situations of limited ventilation and apnoea, the continuous flow of oxygen to the alveoli could maintain the levels of partial pressure of the gas in the blood at adequate levels for a longer time, increasing the safety margin for the patient and a greater time for the anesthesiologist to define strategies to face the situation.

Could the authors discuss the applicability of this approach to children with developmental disabilities undergoing surgery. Thank you for the opportunity. Oxyhemoglobin desaturation is faster in patients with reduced O2 transport capacity, that is, in those with reduced FRC. General anesthesia appears as a major risk factor for mortality in pediatric surgical patients, as well as problems with airway management in patients with comorbidities seems to add life-threatening to this population. In paediatric patients, lung volume and capacity are directly related to weight, height, and age, i.e. the younger and smaller the child, the lower the FRC, this risk being apparently greater in children under three years of age. We can speculate, therefore, that children with developmental disabilities may be at increased risk of hypoxemia during the period of apnea since they have in theory impaired pulmonary oxygen reserves. This work did not direct its scope to the investigation of these patients, however, it is possible to imagine that in a research that applies CPAP to these children under the conditions described here, documenting the effect of this ventilatory mode on FRC, the outcomes could be favorable in relation to the reduction of risk desaturation and prolonging a safe apnea time.

The primary outcome is not clearly described. At first reading, it seems to be the risk of SpO2 of 95% within a 5 minute period (indeed the sample size calculation is that for estimating a risk/proportion). However, reading the manuscript it is repeatedly referred to as hazard of the outcome, implying that the outcome is a time-to-event. The authors need to clarify this. Thank you. We tried to clarify it in the text with some modifications.

The units and resolution of measurement of the outcome T1 is not clear: was it measured down to the second or was. it also measured in 30 second increments? Thank you. The T1 was measured down to the second. Once its record was 95%, we stopped the clock and registered the exact second in which it occurred. During the time elapsed from the start of the apnoea and the 95% threshold, we chose the reference points 10, 20, 30, 40 seconds and so on to record the pulse oximetry value at that moments. Therefore, we recorded SpO2 values every 10 seconds on the data collection form of each patient. For graphic recording purposes, we group these measurements at a 30-second intervals.

If the primary outcome is a risk/proportion, then the sample size calculation is not sufficiently described as it does not include the 'baseline' risk of the outcome; this should be included along with a citation to support it. It is also not clear whether the 50% risk reduction proposed is an absolute or relative reduction. However, if the outcome is time-to-event, then the sample size calculation is incorrect, because other than including the hazard of the outcome in the 'baseline' group, it must also include some determination of the expected number of events. The sample size was calculated using Schoenfeld’s procedure with a sample of at least 33 patients in each randomized group. The study would have a 80% power to detect a 50% reduction in the hazard of the CPAP group, allowing only a 5% chance of a type I error at a two-sided significance test. However, after your observation, we reviewed this calculation and in fact we did not account for the percentage of censored values in the control group. Thus, using the Schoenfeld method, the expected number of events was 66. But supposing that, in the control group, 5% of patients reach the end of the follow-up time with SPO2> 95%, the sample size of each group would be 38 with a significance level of 5% and a power of 80%. On the other hand, if we change only the hazard ratio to 0.45, the sample size for each group would be 30 and the expected number of events would be 50. Please, would be so kind as to help us to find a way out of this issue?

For the descriptive analysis the authors should follow standard reporting for randomised trials, where the mean and SD for continuous baseline variables or counts and proportions for categorical baseline variables be presented by treatment arm without statistical tests comparing the arms. Thank you. Altered in the manuscript

If the primary outcome is a risk within a defined time period (5 minutes) therefore describing it as a hazard and estimating hazard ratios comparing the treatment arms for this outcome is not appropriate. It is however appropriate for the secondary outcome T1 (and perhaps T2). Thank you. We tried to clarify it in the text with some modifications.

The results for the three outcomes should be tabulated in the standard reporting format for a clinical trial i.e. including the event numbers and denominators per group, risks/hazards/rates as appropriate and their ratios, confidence intervals and p-values. Added Table 2.

Please indicate whether the trial protocol submitted with this manuscript was published before the analysis was undertaken, as I am not able to determine that. The trial protocol was published in February 14, 2018, and the data collection begun in March, 2018. The study protocol was then published before the study analysis was undertake.

---

## [Decision Letter · Decision Letter 1]

25 Mar 2021

PONE-D-20-33942R1

Continuous positive airway pressure to reduce the risk of early peripheral oxygen desaturation after onset of apnoea in children: a double-blind randomised controlled trial

PLOS ONE

Dear Dr. dos Santos Neto,

Thank you for submitting your manuscript to PLOS ONE. After careful consideration, we feel that it has merit but does not fully meet PLOS ONE’s publication criteria as it currently stands. Therefore, we invite you to submit a revised version of the manuscript that addresses the points raised during the review process.

As academic editor, I agree with the assessment of the reviewers and I am happy to transmit their Overall positive evaluation. Now, we ask you for carefully addressing their comments, because they can be important for improving your mansucript.

We look forward to receiving your revised manuscript.

Kind regards,

Thomas Penzel

Academic Editor

PLOS ONE

Journal Requirements:

Reviewers' comments:

Reviewer's Responses to Questions

**Comments to the Author**

1. If the authors have adequately addressed your comments raised in a previous round of review and you feel that this manuscript is now acceptable for publication, you may indicate that here to bypass the “Comments to the Author” section, enter your conflict of interest statement in the “Confidential to Editor” section, and submit your "Accept" recommendation.

Reviewer #2: (No Response)

Reviewer #3: All comments have been addressed

Reviewer #4: (No Response)

2. Is the manuscript technically sound, and do the data support the conclusions?

Reviewer #2: Yes

Reviewer #3: Yes

Reviewer #4: Yes

3. Has the statistical analysis been performed appropriately and rigorously? 

Reviewer #2: Yes

Reviewer #3: Yes

Reviewer #4: Yes

4. Have the authors made all data underlying the findings in their manuscript fully available?

Reviewer #2: Yes

Reviewer #3: Yes

Reviewer #4: Yes

5. Is the manuscript presented in an intelligible fashion and written in standard English?

Reviewer #2: Yes

Reviewer #3: Yes

Reviewer #4: No

6. Review Comments to the Author

Reviewer #2: Abstract: in the third-from-bottom line, please report the exact p-value instead of p<0.05 unless it is very small e.g. p<0.001.

Introduction: given your response to the previous round of reviews, it is not correct to say that you measured SpO2 every 30 seconds as indicated in the last paragraph of the introduction; perhaps it is just better to say that you measured time to recovery of pre-apnoea SpO2 levels. (The way this has been described in the paragraph in the methods just above 'statistical analysis' seems fine though.)

Methods:

- similar to the comment above, given your response it is misleading to say in the 'materials and methods' section that you recorded SpO2 values every 30 seconds, if you were continuously checking SpO2 to determine the actual time in seconds to recovery to pre-apnoea levels, unless you were doing this in 30-second increments, which you have indicated in the response that you were not doing.

- indicating the number of children in the first line of materials and methods is actually a non-standard reporting practice, as this is actually a result. It would have been sufficient to say 'we conducted a randomised double blind controlled parallel group trial in children undergoing surgery at...'

- the methods also need to describe the treatment received by the control group. You could add this immediately after describing the intervention, just before where you describe the recruitment in the surgical wards.

- the description of the sample size calculation is still incomplete and not reproducible. Now that it is clearer that this was a time-to-event outcome, for 80% power to detect a 50% relative reduction in hazard you would expect to observe 66 events (this is not the sample size); however your sample size calculation should further indicate what the expected proportions remaining event-free at 300 seconds, given that the sample size would be obtained by dividing the expected number of events by [1 - ((S1+S2)/2)] where S1 and S2 are the proportions in the control and intervention groups respectively that remain event-free at 300 seconds. If under standard care S1 participants remain event free at 300 seconds, then for a relative reduction in hazard of 50%, S2 = S1 x exp(0.5). S1 should have been determined based on information available before this study.

- it is unclear where the chi-squared tests, Fisher's exact test, students t-test etc referred to in the statistical methods were applied. In any case, they don't appear to be necessary at any stage of the analysis.

- what you report in the methods regarding the test for normality is actually a result; instead it would have been sufficient to indicate here that for continuous measures, the test for normality was conducted and you reported means and SDs of normally distributed variables or medians and IQRs for non-normal quantities - without alluding to the results of this test.

Results

- the key to figure 3 doesn't enable the reader to distinguish between the intervention and control group.

Reviewer #3: The authors answered all comments correctly.

I've only some minor comments:

A) Method section: The authors should specify whether children with genetic syndromes, acute and chronic cardiovascular and neurological diseases were included or excluded from the study.

B) page 11: please correct the following sentences:

-"Fig 2 shows the plots of Kaplan-Meier curves for groups CPAP and Control" with "Figure 2 shows the Kaplan-Meier curves for the CPAP and control groups"

-"between-group with between groups"

Reviewer #4: The authors present a revised manuscript on the effects of apneic CPAP after general anesthesia induction in children, concluding that the use of CPAP can prolong time to desaturation. While these results are not surprising, they do highlight that post-induction CPAP may be underutilized in pediatrics and provide a nice literature review in the discussion on pre-intubation CPAP use in pediatric and adult anesthesia.

The article is well-written and has been edited for language and grammar since the prior version, although still contains some grammatical and syntax issues and would benefit from English language editing. I have attempted to highlight some of these below but my suggestions are by no means complete.

The study is well-designed and controlled, appearing to be scientifically sound. The concerns of the prior reviewers seem to have been adequately addressed. Regarding point #9 on power analysis, as any change to the power calculation would be a posteriori I would recommend still reporting the calculations that were performed a priori. If the authors wish to include an additional a posteriori calculation based on the reviewer recommendations to demonstrate the true power of the study, I would suggest stating exactly what happened: the study population size was based on the a priori calculation, but a posteriori a second analysis was performed based on feedback that the initial assumptions required adjustment, and this analysis revealed the study was powered at X% to detect Y difference, or something to that effect.

Specific comments:

Abstract: as a result of deletions, T1 is no longer defined in the abstract, but T2 is. The T1 definition can be appended to the sentence about the primary outcome.

P7 third paragraph: by ‘cardioscopy’ do you mean electrocardiogram?

P7 third paragraph: if 60% fraction was oxygen, was the rest nitrogen gas (room air)? This should be stated.

P8 first paragraph: change ‘halt’ to ‘halted’

P11/Figure 3: why does the control group return to a non-significant difference at 240 seconds? Is this due to patients in the control group getting bag mask ventilation due to falling saturations (post T1 measurement)? If so this figure is misleading and the more appropriate analysis would be to look at only patients who had not yet reached the T1 endpoint when comparing the groups.

P13 second paragraph: change ‘define the airway’ to ‘secure the airway’

P13 last word: change overpasses to surpasses

P14 second paragraph: you state that other studies ‘[show] improvements in outcomes by the use of positive pressure’. Is there definitive evidence that outcomes are improved with CPAP? I can see reduced rates of transient hypoxia during apnea, as you have shown here, but I don’t believe there is evidence that meaningful case outcomes such as postop oxygen requirements, reintubation requirements, etc. are reduced. Thus I would recommend defining what outcomes, if any, are improved.

P14 third paragraph: change ‘being one’ to ‘one being’

P14 third paragraph: change ‘showed’ to ‘shown’

P14 fourth paragraph: change ‘hold the small airways opened’ to ‘prevent atelectasis’

P15 first paragraph: I would change the word ‘threatening’ to something else, maybe ‘the impairment of these factors may predispose patients to harm’ or delete the whole sentence

7. PLOS authors have the option to publish the peer review history of their article (what does this mean?). If published, this will include your full peer review and any attached files.

Reviewer #2: No

Reviewer #3: No

Reviewer #4: No

---

## [Author Response · Author response to Decision Letter 1]

25 Apr 2021

Initially, we are grateful for all the suggestions. Without a doubt, they praised our paper. All questions were answered directly in the original text and we remain at your disposal for any clarifications that are still needed.

1. Abstract: in the third-from-bottom line, please report the exact p-value instead of p<0.05 unless it is very small e.g. p<0.001. Thank you.

Corrected in the manuscript

2. Introduction: given your response to the previous round of reviews, it is not correct to say that you measured SpO2 every 30 seconds as indicated in the last paragraph of the introduction; perhaps it is just better to say that you measured time to recovery of pre-apnoea SpO2 levels. (The way this has been described in the paragraph in the methods just above 'statistical analysis' seems fine though.).

Thank you. Corrected in the manuscript.

Methods:

3. similar to the comment above, given your response it is misleading to say in the 'materials and methods' section that you recorded SpO2 values every 30 seconds, if you were continuously checking SpO2 to determine the actual time in seconds to recovery to pre-apnoea levels, unless you were doing this in 30-second increments, which you have indicated in the response that you were not doing.

Corrected in the manuscript.

4. indicating the number of children in the first line of materials and methods is actually a non-standard reporting practice, as this is actually a result. It would have been sufficient to say 'we conducted a randomised double blind controlled parallel group trial in children undergoing surgery at…'

Thank you. Corrected in the manuscript.

5. the methods also need to describe the treatment received by the control group. You could add this immediately after describing the intervention, just before where you describe the recruitment in the surgical wards.

Added in the manuscript.

6. the description of the sample size calculation is still incomplete and not reproducible. Now that it is clearer that this was a time-to-event outcome, for 80% power to detect a 50% relative reduction in hazard you would expect to observe 66 events (this is not the sample size); however your sample size calculation should further indicate what the expected proportions remaining event-free at 300 seconds, given that the sample size would be obtained by dividing the expected number of events by [1 - ((S1+S2)/2)] where S1 and S2 are the proportions in the control and intervention groups respectively that remain event-free at 300 seconds. If under standard care S1 participants remain event free at 300 seconds, then for a relative reduction in hazard of 50%, S2 = S1 x exp(0.5). S1 should have been determined based on information available before this study.

Thank you for the support. To recalculate the sample size, a 5% survival rate was admitted in the control group at the end of the study. Thus, S1 = 0.05. Therefore, S2= 0.050.5 = sqrt(0.05) = 0.2236 and the sample size would be approximately equal to 76 patients: ([1 - (S1+S2)/2)] = [1 – (0.05 +0.2236)/2] = [1 - 0.2736/2] = 1 - 0.1368 = 0.8632, n = 66/0.8632 = 76.45968). With a sample of 68 patients, the power of the study is 72,4%. During the discussion with our hospital's anaesthesiology clinical staff to limit the observation time to 5 min due to ethical issues, we decided to use data previously obtained in our own unit (pilot study) before patient collection to determine the 5% threshold.

7. it is unclear where the chi-squared tests, Fisher's exact test, students t-test etc referred to in the statistical methods were applied. In any case, they don't appear to be necessary at any stage of the analysis.

Thank you. Corrected in the manuscript.

8. what you report in the methods regarding the test for normality is actually a result; instead it would have been sufficient to indicate here that for continuous measures, the test for normality was conducted and you reported means and SDs of normally distributed variables or medians and IQRs for non-normal quantities - without alluding to the results of this test.

Thank you. Corrected in the manuscript.

Results

9. the key to figure 3 doesn't enable the reader to distinguish between the intervention and control group.

Figure corrected

10. Method section: The authors should specify whether children with genetic syndromes, acute and chronic cardiovascular and neurological diseases were included or excluded from the study.

Thank you. Our population was formed by children with physical status I or II according to the American Society of Anesthesiologists classification. The only four children (3 in the CPAP group and 1 control) with ASA physical status equal to II had controlled asthma. Children with genetic syndromes, acute and chronic cardiovascular and neurological diseases were not included.

page 11: please correct the following sentences:

11. “Fig 2 shows the plots of Kaplan-Meier curves for groups CPAP and Control" with "Figure 2 shows the Kaplan-Meier curves for the CPAP and control groups”

Corrected in the manuscript.

12. ”between-group with between groups”

Corrected in the manuscript.

13. The study is well-designed and controlled, appearing to be scientifically sound. The concerns of the prior reviewers seem to have been adequately addressed. Regarding point #9 on power analysis, as any change to the power calculation would be a posteriori I would recommend still reporting the calculations that were performed a priori. If the authors wish to include an additional a posteriori calculation based on the reviewer recommendations to demonstrate the true power of the study, I would suggest stating exactly what happened: the study population size was based on the a priori calculation, but a posteriori a second analysis was performed based on feedback that the initial assumptions required adjustment, and this analysis revealed the study was powered at X% to detect Y difference, or something to that effect.

Thank you. Corrected in the manuscript.

Specific comments:

14. Abstract: as a result of deletions, T1 is no longer defined in the abstract, but T2 is. The T1 definition can be appended to the sentence about the primary outcome.

Corrected in the manuscript.

15. P7 third paragraph: by ‘cardioscopy’ do you mean electrocardiogram?

Yes, that is correct. Modified in the manuscript.

16. P7 third paragraph: if 60% fraction was oxygen, was the rest nitrogen gas (room air)? This should be stated.

Information added in the manuscript.

17. P8 first paragraph: change ‘halt’ to ‘halted’

Corrected in the manuscript.

18. P11/Figure 3: why does the control group return to a non-significant difference at 240 seconds? Is this due to patients in the control group getting bag mask ventilation due to falling saturations (post T1 measurement)? If so this figure is misleading and the more appropriate analysis would be to look at only patients who had not yet reached the T1 endpoint when comparing the groups.

Thank you. The Figure 3 refers to the measurements made during the T1 fase, after the apnoea onset till the pulse oximetry equals 95%. Once the patient reached a 95% pulse oximetry, the fase T2 fase started and this data is not represented in Figure 3. Our hypothesis is that the lack of significance after 240 seconds is because the groups tend to match up from that moment on, that is, the performance of CPAP would have the greatest impact between 60 and 210 seconds. After 240 seconds, those patients in the control group who did not desaturate would probably not benefit from using CPAP, showing that the benefit of the intervention would be in those patients more likely to desaturate early. 

19. P13 second paragraph: change ‘define the airway’ to ‘secure the airway’

Corrected in the manuscript.

20. P13 last word: change overpasses to surpasses

Corrected in the manuscript.

21. P14 second paragraph: you state that other studies ‘[show] improvements in outcomes by the use of positive pressure’. Is there definitive evidence that outcomes are improved with CPAP? I can see reduced rates of transient hypoxia during apnea, as you have shown here, but I don’t believe there is evidence that meaningful case outcomes such as postop oxygen requirements, reintubation requirements, etc. are reduced. Thus I would recommend defining what outcomes, if any, are improved.

Thank you. Information added in the manuscript.

22. P14 third paragraph: change ‘being one’ to ‘one being’

Corrected in the manuscript.

23. P14 third paragraph: change ‘showed’ to ‘shown’

Corrected in the manuscript.

24. P14 fourth paragraph: change ‘hold the small airways opened’ to ‘prevent atelectasis’

Altered in the manuscript.

P15 first paragraph: I would change the word ‘threatening’ to something else, maybe ‘the impairment of these factors may predispose patients to harm’ or delete the whole sentence

Deleted in the manuscript.

---

## [Decision Letter · Decision Letter 2]

9 Jul 2021

PONE-D-20-33942R2

Continuous positive airway pressure to reduce the risk of early peripheral oxygen desaturation after onset of apnoea in children: a double-blind randomised controlled trial

PLOS ONE

Dear Dr. dos Santos Neto,

Thank you for submitting your manuscript to PLOS ONE. After careful consideration, we feel that it has merit but does not fully meet PLOS ONE’s publication criteria as it currently stands. Therefore, we invite you to submit a revised version of the manuscript that addresses the points raised during the review process.

We are happy with your revision. However some very few, but important comments remain to be answered. We look forward for a good statistical treatment of these concerns.

We look forward to receiving your revised manuscript.

Kind regards,

Thomas Penzel

Academic Editor

PLOS ONE

Journal Requirements:

Reviewers' comments:

Reviewer's Responses to Questions

**Comments to the Author**

1. If the authors have adequately addressed your comments raised in a previous round of review and you feel that this manuscript is now acceptable for publication, you may indicate that here to bypass the “Comments to the Author” section, enter your conflict of interest statement in the “Confidential to Editor” section, and submit your "Accept" recommendation.

Reviewer #2: (No Response)

Reviewer #4: All comments have been addressed

2. Is the manuscript technically sound, and do the data support the conclusions?

Reviewer #2: Yes

Reviewer #4: Yes

3. Has the statistical analysis been performed appropriately and rigorously? 

Reviewer #2: Yes

Reviewer #4: Yes

4. Have the authors made all data underlying the findings in their manuscript fully available?

Reviewer #2: Yes

Reviewer #4: Yes

5. Is the manuscript presented in an intelligible fashion and written in standard English?

Reviewer #2: Yes

Reviewer #4: Yes

6. Review Comments to the Author

Reviewer #2: The authors have responded very well to previous comments. There is still one issue that the authors need to revisit. The sample size calculation is still insufficiently described. First, I would recommend that the authors include a reference to the Schoenfeld’s procedure which they used in the original sample size calculation, and also state the parameters/assumptions that went into the calculation. In the 'revised' calculation which is included in the authors' response to previous comments, the formula is incorrectly applied, specifically where the authors divide S2 by 2 (the correct application is to first sum up S1 and S2 and then divide this sum by 2; the result of this calculation is then subtracted from 1; and finally, the result of this second part of the calculation is multiplied by the expected number of events (i.e. 66 expected events for 50% relative reduction at 80% power) to obtain the overall sample size. Also, please refer to the sample size calculation as such, and not 'study population size'.

In response to previous reviews, the authors removed mentions of chi-squared tests, Fisher's exact test, students t-test etc. Mentions of Shapiro–Wilk test should also be moved. A reference to the Benjamini–Hochberg procedure should also be included, along with a brief explanation of how it adjusts the p-value to avoid type 1 error.

Reviewer #4: (No Response)

7. PLOS authors have the option to publish the peer review history of their article (what does this mean?). If published, this will include your full peer review and any attached files.

Reviewer #2: No

Reviewer #4: No

---

## [Author Response · Author response to Decision Letter 2]

28 Jul 2021

We are grateful for all the suggestions. Without a doubt, they praised our paper. We’ve made some minor adjustments in the text: an excerpt that was previously in the methods was moved to the discussion as we understand that this would be more appropriate since it refers to the limitations of the study, and the reference number 18 was updated. All questions were answered directly in the original text and we remain at your disposal for any clarifications that are still needed.

The authors have responded very well to previous comments. There is still one issue that the authors need to revisit. The sample size calculation is still insufficiently described. First, I would recommend that the authors include a reference to the Schoenfeld’s procedure which they used in the original sample size calculation, and also state the parameters/assumptions that went into the calculation. In the 'revised' calculation which is included in the authors' response to previous comments, the formula is incorrectly applied, specifically where the authors divide S2 by 2 (the correct application is to first sum up S1 and S2 and then divide this sum by 2; the result of this calculation is then subtracted from 1; and finally, the result of this second part of the calculation is multiplied by the expected number of events (i.e. 66 expected events for 50% relative reduction at 80% power) to obtain the overall sample size.

- Thank you for the opportunity. Unfortunately, S2= 0.050.5 was a printing error made by us in the last Response to Reviewers. It was not our intention to and we did not divide S2 by 2. Below we show in detail how the initial and revised sample calculations were performed.

1. Assumptions for original sample size calculations:

1.1 We want to able to detect a 50% reduction in the hazard of the experimental group with a power of 80% and a significance level of 0.05.

1.2 We assumed no censoring.

The reference for Schoenfeld procedure used in original sample size calculation comes from the following Stata’s SE 12.1 command that we used to calculate the sample size:

stpower logrank, hratio(0.5) power(0.8) schoenfeld

Note: By default, stpower assumes an equal allocation design (input parameter: p1 = 0.5000)

Output (original sample size calculation):

2. Revised calculation of sample size:

2.1 We want to able to detect a 50% reduction in the hazard of the experimental group with a power of 80% and a significance level of 0.05.

2.2 We assumed that 5% of the subjects in the control group were expected to survive to end of the study (300 seconds).

Under these assumptions, the required sample size should be:

E = estimated number of events (E = 66 from the output above) and = 1– (S1 + S2)/2. Now,

 S2 = S1HR = 0.050.5 = sqrt(0.05) = 0.2236. So, = 0.8632 and the sample size N ≅ 76. 

This result is confirmed by the following Stata’s stpower output:

stpower logrank 0.05, hratio(0.5) power(0.8) Schoenfeld

Output:

The actual study sample was 68 patients. With this sample size we achieved 72,4% power to detect a 50% reduction in the hazard of the experimental group, with a 5% significance level.

The reference for Schoenfeld’s procedure (formula) used in original sample size calculation is found in An Introduction to Survival Analysis Using Stata, by Mario Cleves et al. Third Edition, 2010, Stata Press, page 338. Here, the author refers to a Schoenfeld’s paper (The Asymptotic Properties of Nonparametric Tests for Comparing Survival Distributions Biometrika 68:316-319) as the source of the formula used in the sample calculations.

This procedure is an option in the following Stata’s SE 12.1 command that we used to calculate the sample size.

Also, please refer to the sample size calculation as such, and not 'study population size’.

- Altered in the manuscript

In response to previous reviews, the authors removed mentions of chi-squared tests, Fisher's exact test, students t-test etc. Mentions of Shapiro–Wilk test should also be moved.

- Altered in the manuscript

A reference to the Benjamini–Hochberg procedure should also be included, along with a brief explanation of how it adjusts the p-value to avoid type 1 error.

-A reference to the Benjamini–Hochberg procedure could be their original article: Benjamini, Y., and Hochberg, Y. (1995). Controlling the false discovery rate: a practical and powerful approach to multiple testing. Journal of the Royal Statistical Society Series B, 57, 289–300. Briefly, this method is one of several for dealing with the so-called multiplicity of hypothesis tests problem (http://www.biostathandbook.com/multiplecomparisons.html). When there are several hypotheses to be tested with the data from a particular study, the problem of the occurrence of false positives (type I error) arises. The criticism of many more traditional methods is that they greatly diminish the chance of rejecting a null hypothesis. This also leads to an unwanted increase in false negatives. Benjamini-Hochberg's proposal is not to try to reduce the chance of a false negative so drastically, but to accept a certain proportion of false positives. From there, it defines a procedure to define which null hypotheses among the rejected ones should really be rejected (true positives). An explanation of how it adjusts the p-value to avoid type 1 error can be find at https://www.statisticshowto.datasciencecentral.com/benjamini-hochberg-procedure/.

---

## [Editor Report · Decision Letter 3]

20 Aug 2021

Continuous positive airway pressure to reduce the risk of early peripheral oxygen desaturation after onset of apnoea in children: a double-blind randomised controlled trial

PONE-D-20-33942R3

Dear Dr. dos Santos Neto,

We’re pleased to inform you that your manuscript has been judged scientifically suitable for publication and will be formally accepted for publication once it meets all outstanding technical requirements.

Kind regards,

Thomas Penzel

Academic Editor

PLOS ONE
---

## [Editor Report · Acceptance letter]

23 Sep 2021

PONE-D-20-33942R3 

Continuous positive airway pressure to reduce the risk of early peripheral oxygen desaturation after onset of apnoea in children: a double-blind randomised controlled trial 

Dear Dr. dos Santos Neto:

I'm pleased to inform you that your manuscript has been deemed suitable for publication in PLOS ONE. Congratulations! Your manuscript is now with our production department. 

Kind regards, 

on behalf of

Dr. Thomas Penzel 

Academic Editor

PLOS ONE